



# The effect of Three Gorges Dam and rainfall on summer
# flow risk over Yangtze River Basin
Zhenkuan Su[1], Zhenchun Hao[1], Michelle Ho[2], Upmanu Lall[2,3], Xun Sun[2,4], Xi Chen[5],
Longzeng Yan[6]
[1] State Key Laboratory of Hydrology-Water Resources and Hydraulic Engineering, Hohai University,
Nanjing 210098, China
[2] Columbia Water Center, Columbia University, New York, NY 10027, USA
[3] Department of Earth and Environmental Engineering, Columbia University, New York, NY 10027, USA
[4] Key Laboratory of Geographic Information Science (Ministry of Education), East China Normal University,
Shanghai 200241, China
[5] Bureau of Hydrology, Changjiang Water Resources Commission, Wuhan 430010, China
[6] Upper Changjiang River Bureau of Hydrology and Water Resources Survey, Chongqing 400020, China
*Correspondence to*: Zhenkuan Su (zhenkuan.su@gmail.com)
**Abstract.** As the largest water conservancy project, Three Gorges Dam starts its impoundment in 2003 and
henceforth the efficient operation of a multi-purpose dam has aroused a great concern on the effectiveness
on flood control and water management over Yangtze River Basin. In this paper, we consider the relationship
between rainfall from 136 weather stations and streamflow from 5 hydrological stations including Cuntan,
Yichang, Luoshan, Hankou and Datong. Meanwhile, the spatial average rainfall over 21 subbasins was
computed. The analysis of the correlation demonstrated that the correlation of spatial average rainfall and
streamflow for each station is consistent with that between rainfall by stations and streamflow. Then, two
options were selected to develop the linear models, including option a) using rainfall by stations to forecast
streamflow and option b) using spatial average rainfall to forecast streamflow. The canonical correlation
analysis enabled a large degree of spatially coherent information of rainfall by linear transforms to maximize
the correlation of rainfall and streamflow for developing linear models. The model resulting from option b)





best fits the observations. Coefficient of determination for each model and statistics such as reduction of error,
coefficient of efficiency under cross validation serve as a test of good performance of the model. An analysis
of streamflow characteristics across the stations illustrated that the streamflow of Cuntan and Yichang have
different modes of variability from that of the rest stations below the dam. The comparison of risk type
changes between observations and predictions during the post-dam period showed the dam effectively
relieved the risk of upstream while not always played a positive role in risk mitigation of downstream. The
spatial and temporal patterns of rainfall anomalies over the whole basin helps to explain how the dam impact
the streamflow. Some applications as to possible operational strategies are discussed.
**Keywords:** Three Gorges Dam, canonical correlation analysis, flood, drought, risk
**1. Introduction**
The Yangtze River, also known as Changjiang in China, is the longest river in Asia and plays a large role in
history, culture and economy of China (Figure 1). Its river basin is home to one-third of the country's people.
The whole drainage basin covers an area of 1.8 million km$^2$ and is comprised of a range of complex terrains
from plateaus to plains. The basin's climate is characterized by complex regional subbasin scale patterns of
precipitation and temperature. The combination of complexity in both climate and terrain results in widely
varying rainfall-runoff process across the basin (Xu et al., 2008). Periodic floods and droughts in rainy
seasons have been experienced in the basin throughout history and have significantly impacted those who
live there and local economy (Heng and Xu, 1999;Wang et al., 2011;Bing et al., 2012;Zhang et al.,
2007;Zhang and Zhou, 2015;Zong and Chen, 2000). For example, 4 970 000 houses were flattened in the
flood of year 1998 and agricultural disaster area was 33 900 km$^2$ over upstream in the drought of year 2006.
In response to these events, the Three Gorges Dam (TGD) was planned and began construction in 1993 with
water storage commencing in 2003. The primary function of what is now the largest project in the world is
to provide flood control and regulate seasonal variability in streamflow to facilitate water allocation (Gleick,
2009). With the dam in operation since 2003, streamflow from flood seasons are stored and released for
hydropower and water allocations during the dry seasons. In this way, the dam modulates the streamflow





distribution over the year (Wang et al., 2013a). Past studies have focused on how the dam has performed in
relation to flood control in addition to other impacts such as water pollution and ecological problem (Wang
et al., 2013a;Gao et al., 2013;Ma et al., 2016;Li and Qiu, 2016). Both the basin-wide drought in 2006 (Dai et
al., 2011;Dai et al., 2012) and the 2016 flood have raised concerns regarding the effectiveness of the dam.
Consequently, analyses of the dam impact on the streamflows are important.
Hydrological models such as the physically based Soil and Water Assessment Tool (SWAT) model and
Variable Infiltration Capacity (VIC) model have been widely used in the analyses of streamflow modelling
and their predictions (Zhang et al., 2012a;Kim et al., 2016;Liang et al., 1994;Yuan et al., 2004;Gayathri et
al., 2015). The models are used for modelling both gauged and ungauged catchments, and help inform flood
control operations, water resources management and climate induced runoff forecasting. However,
distributed hydrological models such as MIKE SHE model and SWAT model are limited by aspects such as
large data requirements, unclear parameter uncertainty and inaccurate physical process (Gayathri et al.,
2015;Pechlivanidis et al., 2011). Moreover, the uneven distribution of rainfall stations and the complex
climate and terrain conditions in Yangtze River Basin make it difficult for a single model to accurately
characterize streamflow across the whole basin.
In contrast, statistically-based models, such as regression and correlation models, use observation-oriented
methods and seek to find the functional relationship between explanatory and response variables(Elsanabary
and Gan, 2015;Kwon et al., 2009;Lima and Lall, 2010;Chen et al., 2014;Steinschneider et al., 2016). These
models are data orientated and do not take into the consideration process driving the hydrological system and
typically comprised of a small amount of model parameters. The simplicity of these models thus allows one
to apply relatively less variables to the streamflow forecasting.
Ultimately, the success of both distributed hydrological models and statistical models depends critically on
the availability of data. In Yangtze River Basin, the streamflow is strongly related to rainfall. For example,
the correlation coefficient between annual precipitation and streamflow at Datong station is 0.85 and the
runoff coefficient is 45% (Zhang et al., 2011). But the use of highly correlated rainfall records among the
adjacent gauges in a single model is problematic. From this point, we will develop a model considering the





inter-site correlations for forecasting seasonal streamflow at multiple sites, which is important for river
system applications.
Flood and drought events of streamflow are typically characterized by above and below normal streamflow.
The economic and environmental consequences of these events are among the most serious of all natural
disasters. As the dam impounded water since 2003, the streamflow shows a different tendency compared to
the records during the pre-dam period.
The objectives of the analyses presented in this paper are:
1) Investigate the relationship between rainfall and streamflow in flood seasons, and analyze the
characteristics of streamflow over the whole basin and their relationships;
2) Develop a plausible statistical approach to model the streamflow across the Yangtze River Basin,
considering the high correlations among the intensive rainfall stations as well as the correlations among the
streamflow stations.
3) Evaluate the current dam operation for risk mitigation and guide future dam operation strategy.
**2. Study area and data description**
To perform the basin-scale risk analysis, we selected streamflow data from five hydrological stations
including Cuntan, Yichang, Luoshan, Hankou and Datong, along the main stream (Figure 1). Cuntan station
is above the TGD is considered to be the starting point of the reservoir formation, while all other gauges are
located downstream of the dam. The Yichang station is known as the "Gateway to the Three Gorges" and is
located approximately 40 km downstream of the dam and 1 840 km from the estuary. Luoshan is located
close to the outlet of the Dongting Lake, the majority of which is located in the Yangtze floodplain, resulting
in the lake's inundation level impacts with the gauge's level. Hankou station provides critical streamflow
monitoring for managing flood control operations given its location downstream of where the Hanjiang River
join the main stem of the Yangtze River. For Datong station, the water level is impacted by the inflow from
the main stream and the upper Poyang Lake. It is also the gauging station used to assess environmental flows
and monitor outflow to the sea. The whole basin is divided into 21 subbasins based on the flow direction and



elevation, and the boundaries are obtained from the water authorities of Changing Water Resources
Commission. Their locations have been shown in the Figure 1.
The daily streamflow records cover the period from 1960 to 2014 for each station. This data was also obtained
from the water authorities of Changjiang Water Resources Commission and have been checked and quality
controlled. Daily streamflow was aggregated into monthly streamflow data using the arithmetic mean. The
annual hydrograph shows the seasonality of streamflow (Figure 2), which is monsoon summer (June-July-
August) flow dominating.
The daily rainfall dataset is accessed from the National Meteorological Information Center
(http://data.cma.cn). Data at 136 weather stations spans from 1960 to 2014 with less than 5% missing data,
which are linearly interpolated using data from the closest two stations. And, the subbasin rainfall is also used
in this paper. The rainfall over a given subbasin is the spatially averaged value at all rainfall stations within
the subbasin and computed by the Thiessen method. This method assigns an area called Thiessen polygon to
each rainfall station. The Thiessen polygon of a rainfall station is the region for which if we randomly choose
any point in the polygon, that point is closer to this specific station than to any other stations. Thus, the rainfall
recorded by each station should be weighted according to the polygon, it represents. Finally, we consider the
summer JJA average streamflow and the sum of rainfall during the same period as variables for detecting
summer risk changes over Yangtze River Basin.
**3. Predictive models for the summer flow**
In this section, summer (JJA) streamflow and rainfall were used to estimate the rainfall-runoff relationship
through the use of canonical variables of canonical correlation analysis, and predictive models were designed
using this relationship. The date of June 2003 is used to delineate the change from a free flowing river to an
impounded river as this was the time at which storage in the reservoir behind the TGD began to accumulate.
The analysis is therefore conducted using two time periods of pre and post June 2003 to assess the impact of
the dam on runoff. The models are fit to the data prior to 2003 to quantify the rainfall-runoff relationship
prior to major streamflow regulations. The basics of canonical correlation analysis are introduced in section





3.2 and the performance metrics are summarized in section 3.3.
**3.1 Model design**
We try to use both rainfall data over stations or spatial average rainfall over subbasins to model the average
streamflow in summer by developing a log-linear model. The summer JJA flow was considered as dependent
variable $y_{t,s}$ in year $t$ at station $s$, and logarithmically transformed as summer streamflow at five stations
were approximately log-normally distributed. Rainfall from 136 weather stations and spatial average rainfall
over 21 subbasins were considered as the independent variables $x_{t,k}$ in year $t$ over station or subbasin $k$.
Initial diagnostic analyses were conducted using the Pearson correlation analysis between the rainfall $x_{t,k}$
and the log transformed streamflow $\ln(y_{t,s})$ (shown in Figure 3). Correlations between streamflow $\ln(y_{t,s})$
and the rainfall $x_{t,k}$ across the whole basin show that the regions with a strong significantly correlation
using spatial average rainfall are consistent with those using data by stations. It reflects the ability of both the
station by station and the subbasin by subbasin regression methods used for rainfall to preserve local climate
details in the streamflow modelling.
In terms of spatial distribution of the correlation map, the values vary over different parts of the whole basin.
For the streamflow of Cuntan station, the most significantly correlated subbasins are Jinshajiangupper,
Yalongjiang and Jialingjiang. These subbasins are the headwater catchments of the Yangtze River and feed
into streamflow at the Cuntan gauge. However, the rainfall in Subbasin Xiangjiang located at downstream of
the dam has negative correlation with the streamflow, indicating that different climate events occurred over
northwestern and southeastern part of Yangtze River Basin. For other streamflow stations, details can be
observed in Figure 3 with blue denoting high positive correlation and red denoting high negative correlation.
The preliminary results indicated that streamflow is highly correlated with both subbasin-scale rainfall and
rainfall at specific locations. As a result, two predictor data sets were used to model streamflow as follows:
**a)** streamflow at each gauge (Cuntan, Yichang, Luoshan, Hankou and Datong) is modelled using gauged
rainfall data from 136 weather stations as predictors in a regularized canonical correlation analysis;
**b)** streamflow at each gauge is modelled using the spatially averaged subbasin rainfall as predictors to model
the streamflow in a canonical correlation analysis (without regularization).





The application of the canonical correlation analysis will be discussed in next section 3.2.
**3.2 Canonical correlation analysis**
In this paper, due to the adjacent locations for the rainfall stations and subbasins, the observed rainfall $x_{t,k}$
from the weather stations or subbasins do interact with each other, and actually the mutual correlation is very
high, even greater than 0.9 for many stations or subbasins. To solve this problem, we implement canonical
correlation analysis (CCA) (Hotelling, 1936) to process the dimension reduction since the rainfall records
over stations have high spatial correlation. Unlike the methods such as principle component analysis (PCA)
(Jolliffe, 2002) and archetype analysis (AA) (Cutler and Breiman, 1994;Steinschneider and Lall, 2015;Stone
and Cutler, 1996) that are usually used for dimension reduction, CCA takes into account the relationship
between the explanatory and target variables. Consequently, we employed CCA to maximize the correlation
between $\ln(y_{t,s})$ and $x_{t,k}$.
In CCA, we consider two data sets (i.e., rainfall $x_{t,k}$ and streamflow $\ln(y_{t,s})$), where $x_{t,k}$ is the
explanatory variable $X$ and the $\ln(y_{t,s})$ is the target variable $Y$. CCA rotates the two sets of variables to
achieve two vectors: $a$ and $b$, such that the random variables $U = a'X$ and $V = b'Y$ maximize the
correlation between the rainfall and streamflow. $U$ and $V$ are then regarded as the first pair of canonical
variables. Next, one seeks vectors maximizing the same correlation subject to the constraint that they are to
be uncorrelated with the first pair of canonical variables. This gives the second pair of canonical variables.
This procedure will be continued up to the minimum value between the number of explanatory variables and
the number of target variables. We can understand in one way that the CCA transfers the multiple variables
to less variables, so that we can use the correlation between the less variables to represent the relationship of
the explanatory variables and target variables.
However, we employ regularized CCA to address the high dimensionality of rainfall variables in the case
where station data are used to prevent overfitting. Regularization is a smoothing process in order to prevent
overfitting and is akin to ridge regression (De Bie and De Moor, 2003). It is typically represented by the
regularization parameter (lambda), which ranges from 0 to 1 with larger lambda indicating a higher degree
of smoothing. The degree of freedom (as defined by Dijkstra (2014)) is limited to a maximum value of $n$-10,



where $n$ is the total number of observations. If the maximum value could not be achieved, lambda was set to
one, otherwise lambda was evaluated to two significant figures. Both regularized CCA and non-regularized
CCA are executed in R using the package "CCA" by Gonzalez et al. (2008).
Here, we got the first 5 pairs of canonical variables, which were used to fit the model of streamflow. For each
pair, we can develop one linear regression equation using the linear model as following Eq. (1). Hence, a
system of equations can be developed and this enables one to use rainfall to predict streamflow by solving it.
$$V_i = \beta_i U_i + \alpha_i + \varepsilon_i \tag{1}$$
where, $i$ is order of the 5 pairs of canonical variables ranging from 1 to 5. $\beta_i$ is the regression coefficient
and $\alpha_i$ is the intercept in $i$-th linear model. $\varepsilon_i$ is the error term.
**3.3 Model performance metrics**
Cross validation is used to verify the predictability of a model against overfitting. Here, we applied a leave-
$m$-out cross validation method, which is processed by leaving out $m$ randomly selected data from the
observations for validation and the remaining $n$-$m$ data are used for model calibration, where $n$ is the total
number of observation. This procedure was be repeated 100 times, resulting in a set of validation metrics
from all the models. In this study, we calculate two metrics to verify the performance and predictability of a
model. These are the reduction of error (RE) and coefficient of efficiency (CE) (Cook et al., 1994;Wilson et
al., 2010) and are similar to the Nash-Sutcliffe efficiency test. RE and CE are defined as following Eq. (2)
and Eq. (3):
$$RE = 1 - \frac{\sum(x_i - \hat{x}_i)^2}{\sum(x_i - \bar{x}_c)^2} \tag{2}$$
$$CE = 1 - \frac{\sum(x_i - \hat{x}_i)^2}{\sum(x_i - \bar{x}_v)^2} \tag{3}$$
Where, $x_i$ is the observation in year $i$ in the validation period and $\hat{x}_i$ is the associated predicted streamflow
data by the model. $\bar{x}_c$ and $\bar{x}_v$ are the mean of observations used for calibration and validation, respectively.
The theoretical limits for the values of both statistics range from a maximum of 1 to minus infinity, with
larger value indicating better prediction skill. A positive value indicates that the model skill is better than
using the climatology of the verification period. Some application in the streamflow modelling context could



be found in (Chen et al., 2014;Devineni et al., 2013;Ho et al., 2016).
**4. Results**
Two log-linear models were developed for predicting the summer streamflow at each of the 5 hydrological
stations using the canonical variates derived from the predictors stipulated by option a) gauged rainfall and
b) subbasin scale rainfall. A cursory comparison of the developed models was firstly performed by comparing
the coefficient of determination for each model and then the cross-validation statistics computed over
different blocks of data sets identify the best fitting model of the streamflow. Finally, risk change in summer
streamflow was analyzed and the spatial variability of the rainfall over the whole basin during the post-dam
period was explored to help explain the risk change.
**4.1 Model results**
Averaged flow in summer at each station along the main stream of Yangtze River was modelled using the
canonical variables from option a) and b), respectively. The modelled results with $5^{th}$-$95^{th}$ prediction interval
have been shown in Figure 4. The development of CCA linear models for each hydrological station either
using option a) or option b) resulted in acceptable models of streamflow, and the mean adjusted $R^2$ for the
two options are 1.0 and 0.80, respectively. Furthermore, the modelled streamflows and observations have a
strong correlation with average values of 1.0 and 0.92 for option a) and option b), respectively, which indicate
that the models are powerful to model and predict the streamflow.
While for the model from option a), the $R^2$ is close to 1, which seems that the modelled streamflows may
overfit the observations. In fact, the spatial average rainfall over a given subbasin can represent the regional
effect from a climate event and the corresponding model was superior to the model from option a), while a
single rainfall station may misrepresent the amount of rainfall received by the watershed. Several researches
have shown an improvement of rainfall-runoff models contributed by spatially distribution comes from
accounting for rainfall variability (Andreassian et al., 2004;Tramblay et al., 2011). To verify this point, a
detailed description of the cross validation for the model results will be further discussed and evaluate the





models in next section 4.2.
**4.2 Cross validation**
For the cross validation, we randomly draw 5-yr data (about 10% of the total number of observations) without
replacement for validation from the time series and the remaining 38-yr data were used to develop a new
model for estimating the regression parameters. This procedure was repeated 100 times to calculate the
validation metrics of RE and CE. The RE and CE values were calculated for option a) and option b),
respectively, and the results are shown in boxplots (Figure 5).
For option a), the median CE value for Yichang is slightly less than zero, indicating that the model predictions
are less skillful than the climatology of the verification period. The median values of RE for Yichang and CE
for Hankou and Datong are close to zero. For option b), we can observe that both the median RE and CE
values for all streamflow sites are greater than zero, indicating that the model from option b) has better
predictive skill than using climatology of the verification period. It reflects that the log-linear model using
spatial average rainfall over subbasins without overfitting is better at predicting the summer flow with greater
accuracy than that using rainfall over stations through an overall assessment. We therefore select the model
from option b) (CCA using spatial average rainfall over subbasins) for further analysis. In next section, we
use predictions of the streamflow during the post-dam period to represent the natural flow without the dam
impact for risk change analysis.
**4.3 Risk analysis**
Summer average streamflows at the five selected gauges within the Yangtze River Basin were modelled using
subbasin scale rainfall. The modelled streamflow represents naturalized flows and was separated into the pre
and post dam periods to assess potential changes in flood and drought risk. Quantiles of summer flow after
2003 relative to the streamflow during the pre-dam period was then used to help estimate the risk with which
floods or droughts of certain magnitude may occur. Furthermore, the spatial patterns of rainfall over the
whole basin help to interpret and analyze how the dam impact the streamflow for the several stations.
**4.3.1 Streamflow characteristics over the whole basin**





Both modelled and observed summer streamflow for the period 1960-2014 are shown in log space in Figure
6. The modelled streamflows closely match the observed streamflows in the period prior to dam operation.
In contrast, the modelled streamflows are noticeable different from the regulated streamflows after 2003,
indicating the dam regulates summer flows.
PCA was used to extract the main modes of variability of modelled natural log streamflows across all 5
stations. A scree test was used to inform the number of PCs to retain (the plot is not shown). The first two
PCs were retained that together explain over 95% of the variation (PCs shown in Figure 7).
For PC 1, the same sign in the loading values across all inputs is basically showing a regionally cohesive
signal. Both the flood in 1998 and drought in 2006 can be explained through the temporal variability of PC
scores. The Yangtze River Basin experienced approximately twice the seasonal average rainfall during the
summer of 1998 resulting in widespread flooding. This event is evident in PC 1. While the severe drought in
more than a century struck southwest China and Sichuan Province in the summer of 2006 and water level of
Yangtze was the lowest in the past century, we can see a large negative PC score in 2006.
PC 2 mainly represents the difference in the upper and lower reach of Yangtze River Basin, and largely
impacts the streamflow across the 5 stations by the regional climate events. Unlike in PC 1, there shows
different loadings for the 5 stations, with positive values for Cuntan, Yichang and negative values for the rest
stations locating in the middle and lower reach of Yangtze River. In 1969, the streamflow anomalies from
upper to lower reach increased while the inverse trend was found in 1981. During the mid-1990s, from Cuntan
to Datong station, the streamflows increased and this was detected in PC 2 with negative values.
**4.3.2 Risk changes of summer flow during the post-dam period**
To quantitatively assess the difference in temporal modes of variability implied by the modelled and observed
streamflow during the post-dam period, the probabilistic analysis is typically used. However, probabilistic
analyses require suitably long records to fit distributions. As an alternative, quantiles of the streamflows were
used to compare the modelled and observed streamflows in the pre and post dam periods at each streamflow
location. Furthermore, we also define a flood occurrence as an event during which the quantile of streamflow
is greater than the 75[th] percentile JJA flow of the pre-dam period, while a drought occurrence is less than the



25th percentile JJA flow of the pre-dam period. The threshold lines were plotted in Figure 8 with dashed line
denoting the flood and black line denoting the drought.
The occurrence of flood and drought in the post dam period were assessed using the modelled streamflows,
to represent theoretical flood and drought occurrences without the dam, and observed streamflows, to
quantify the flood and drought mitigation capacities enabled by dam operations. These results are
summarized in Figure 8 and Table 1. We can see that the dam mitigates the flood risk for Cuntan and Yichang
over the upper reach purely considering the number of years (e.g., 2003, 2010, 2013). In contrast, the
occurrence of floods and droughts are not noticeably alleviated by dam operations in the middle and lower
reaches as these streamflows are impacted by Dongting Lake and Poyang Lake.
The dam influence may be qualitatively assessed by considering the spatial pattern of subbasin rainfall
anomalies and observed streamflows in the period after 2003 (rainfall anomalies map shown in Figure 9). In
both 2003 and 2010, anomalously high rainfalls were experienced in the subbasins upstream (e.g.,
Jialingjiang) of the Cuntan and Yichang gauges resulting in flood conditions. However, flood impacts were
alleviated through dam operations with lower observed flows at the Yichang gauge located downstream of
the dam relative to the modelled flows. Decreased flows were also observed at the Cuntan gauge upstream
of the dam in 2003, likely a result of reservoir backwater as the reservoir filled. However, streamflow at the
remaining three gauges further downstream were not notably impacted by dam operations.
In 2008, the rainfall over the upstream portion of the basin were close to the average while lower basin
experienced above average rainfall. What is worse is that a spurious "drought" is observed at Cuntan as the
gauge was impounded by the reservoir and this resulted in that the backwater blocked the flow and reduced
the flow velocity. Despite the above average rainfalls in the lower basin, reduced outflows from the dam as
a result of average inflows into the dam meant that the region downstream of the dam was in an effective
drought.
In 2012 and 2013, the rainfall over the upstream is approximately average, and the dam operations do not
cause large changes in hydrological risk for Cuntan and Yichang. Conversely, negative rain anomalies over
some subbasins surrounding the two lakes largely impacted the outflow from the lakes into the main Yangtze

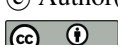



stream in 2013. The abnormal rainfall and river-lake interaction resulted in smaller values of observations.
This was also detected in 2012 but not obvious in the spatial pattern.
**5. Conclusions and discussions**
Human activities and climate change irrefutably have a profound influence on the hydrologic cycle and
intertwine with each influencing and affecting the behavior of the other (Wang et al., 2013b). This study
examined the potential to use the basin-scale rainfall to directly model the streamflow and evaluate the effect
of dam operations on summer flow risk. We considered two options in the model designing, which included
the model using rainfall by stations and the model using spatially averaged rainfall by subbasins. Instead of
the distributed physically-based models, a single statistical model was developed and it used relatively less
variables to model streamflows but considered the spatial correlation of streamflow over the whole basin.
The CCA approach enabled a large degree of spatially coherent information of rainfall by linear transforms,
with which the correlated rainfall and streamflow could be expressed using linear equations. The best fitting
model was selected to be the model using spatial average rainfall over subbasins and cross-validation method
served as a test of better performance with the result consistent with the expectations from fitting. This
method can be extended to apply in the area where less available data was obtained and each independent
variable has a high correlation with others.
The result of our study presented that, since 2003, there has been a certain degree of difference between the
observations and the natural flows without dam running. Floods and droughts experienced in the post dam
period were amplified, driven, or alleviated. In summer, the rainfall is the headwater of Yangtze's streamflow
and the dam's behaviors are typically taken forcedly by the operators. Therefore, the dam regulation has the
most significantly impact on the streamflow, compared to other driven factors such as land use change and
suspended sediment. The analysis of variance of streamflow explained by each predictor could be the next
research topic.
Changes in discharge and water level of the river change the blocking force of the river on the outflows from
many large hydrological entities along the Yangtze River such as Dongting Lake, Poyang Lake and the key



tributaries (Guo et al., 2012;Zhang et al., 2012b;Dai et al., 2016;Lai et al., 2014). The TGD changes the
Yangtze River discharge and water level and subsequently disturb the interrelations of the river and lakes,
and affects the river flow and hydrologic cycle of the lakes. Similarly, in Zhan et al. (2015)'s study, they took
triplicate samples of the river and Dongting Lake to determine the relationship between river-lake interaction,
and found that during 3 different dispatching periods including water-supply, flood-storage and water-storage
dispatching period, the lake is recharged through different rivers. Through the dam operation, it should be
ensured that the water level and discharge over both upstream and downstream are not too low in dry seasons
and not too high in wet seasons according to the rainfall distribution over the whole basin, and thus prevent
the floods and droughts.
In this paper, we used the rainfall to model the streamflow and this method could also be applied to any
regional areas where are not easy to directly relate the pre-season climate and streamflows. The approach
demonstrated here enabled one to develop and test both the rainfall induced variations and changes due to
the human activities on a river. Much research is also directed towards an analysis of the river-lake interaction
in a changing spatial pattern of climate.
**Acknowledgements**
We would like to sincerely thank the editors and reviewers who provided valuable comments, Changjiang
Water Resources Commission for providing the streamflow data, National Meteorological Information
Center (http://data.cma.cn) for providing the rainfall dataset, our colleagues working at Hohai University and
Columbia Water Center of Columbia University for going over the result. This work was partly supported by
China Scholarship Council. It was also funded by the National Key Research Projects (Grant NO.
2016YFC0402704) and National Natural Science Foundation of China (Grant NO. 41371047).

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





**Table 1: The years when the risk type changed during the post-dam period. Three cases are considered, which**
**include Case 1: "None→Risk" denotes hydrological extremes from none to having; Case 2: "Risk→None" denotes**
**hydrological extremes from having to none; Case 3: "Risk→Opposite Risk" denotes hydrological extremes**
**changed from flood to drought or from drought to flood.**

| Stations | None→Risk | Risk→None | Risk→Opposite Risk |
|---|---|---|---|
| Cuntan | 08 | 03,10,13 | 14 |
| Yichang | 08 | 03,10 | 14 |
| Luoshan | 08,12,13 | 04,14 | |
| Hankou | 12,13 | | 08 |
| Datong | 12 | 07 | 08,13 |








**Figure 1: The locations of the rainfall stations, hydrological stations and subbasins used in the analyses, the Yangtze River Basin shown in grey in the bottom left. The subbasins are annotated with number and names are shown in the top of this figure.**






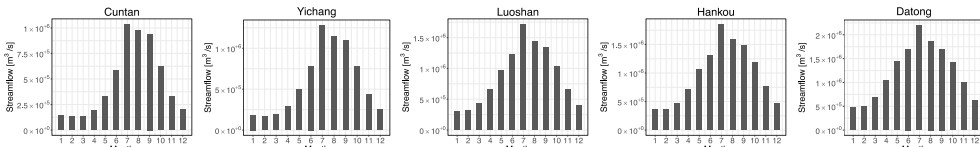


**Figure 2: Monthly average streamflow for each station over Yangtze River Basin.**




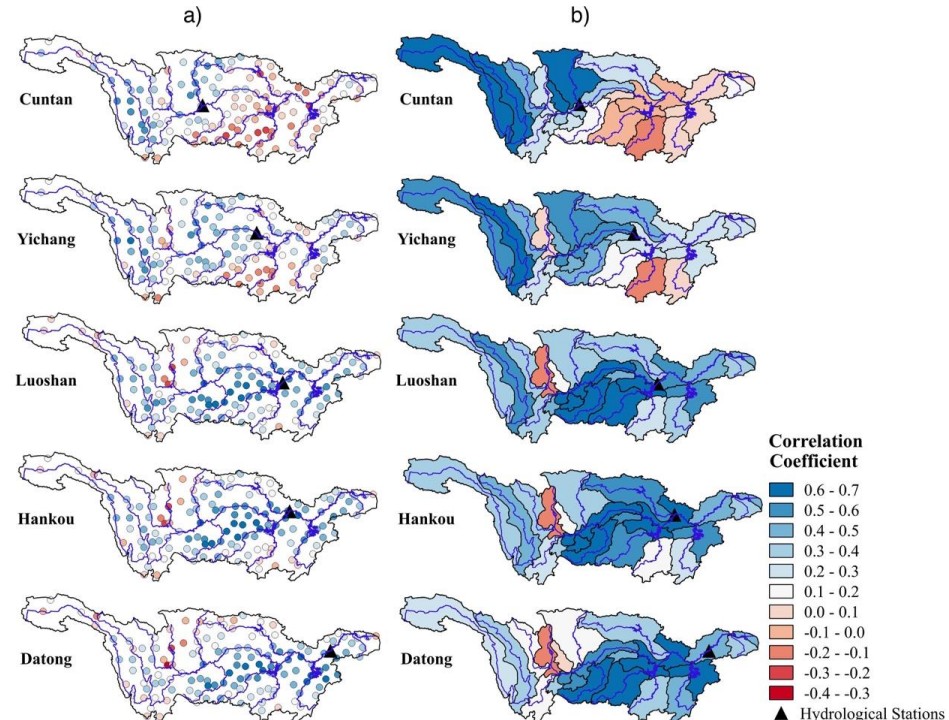


**Figure 3: Pearson correlation map between the average streamflow in summer (JJA) for each hydrological station**
**and a) rainfall over stations, b) spatial average rainfall over subbasins in summer (JJA).**

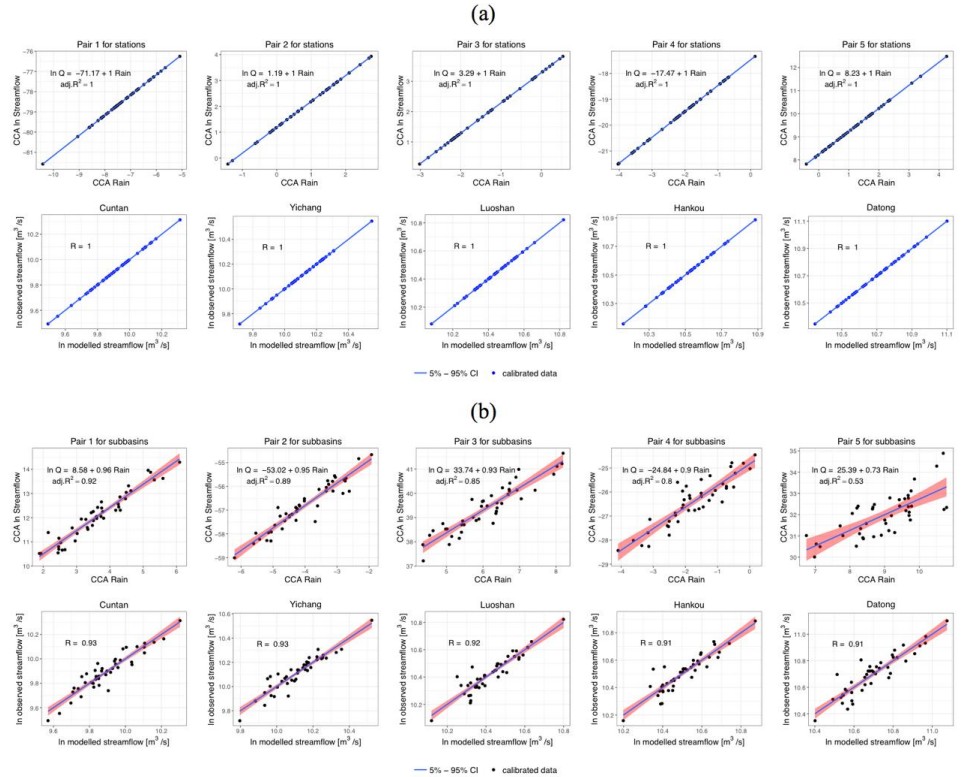

**Figure 4: Model results with 5th-95th prediction interval for the streamflows over the Yangtze River Basin calibrated using available data (1960-2002) and the canonical variate of rainfall by stations (a) and by subbasins (b), respectively. The first row in both (a) and (b) is the result of developed models using 5 pairs of canonical variables, and second row is the result of observed and modelled streamflow for each hydrological station.**




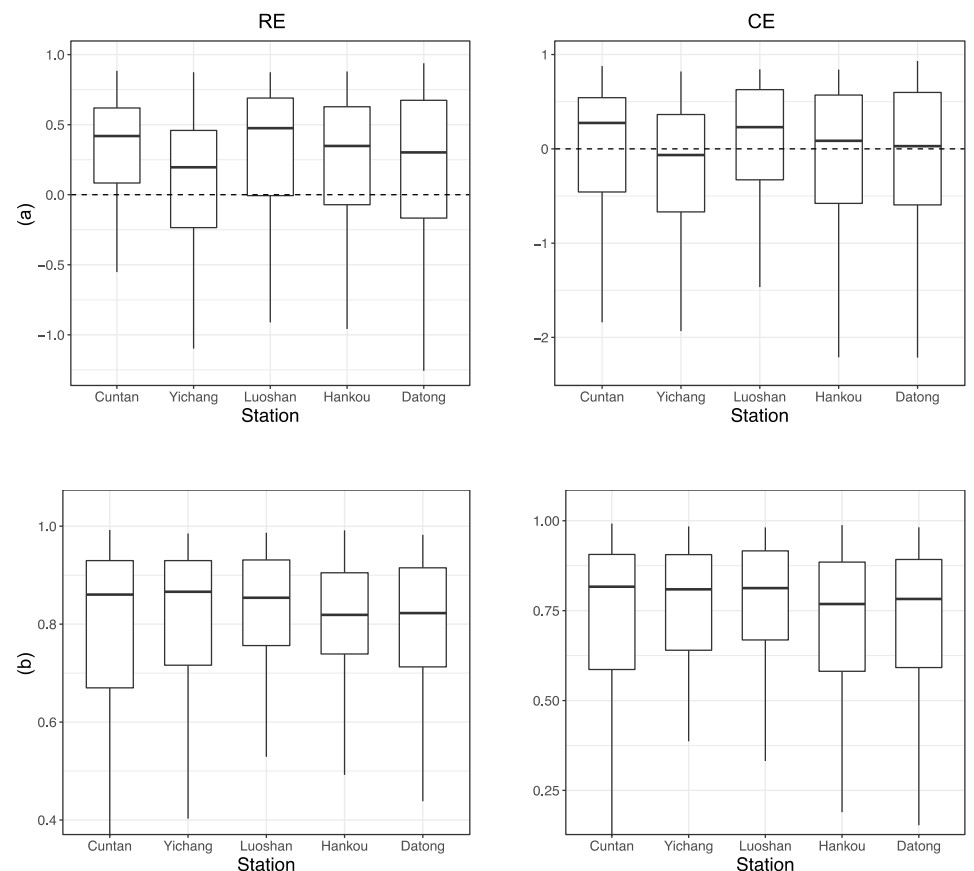


**Figure 5: Boxplots of RE and CE under cross validation. (a) and (b) on the left side indicate the two different**


**predictor data sets from option a) and b), respectively.**








**Figure 6: Model results and observations with 5-yr moving average of the streamflow over Yangtze River Basin.**

**The dashed line is the average streamflow for each station during pre-dam period.**






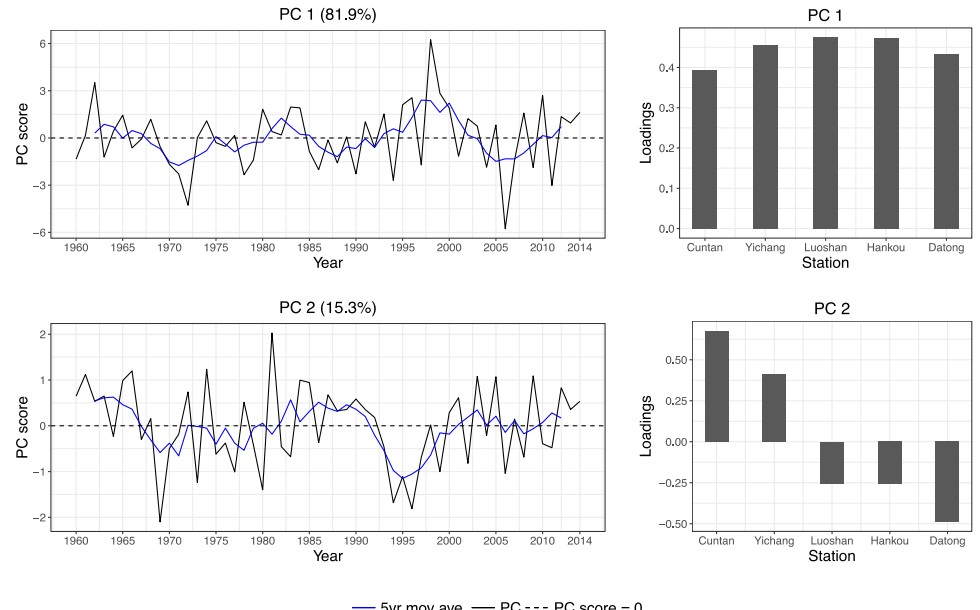


**Figure 7: The first two PCs of summer average streamflows across the 5 hydrological stations and their 5-yr**
**moving average over Yangtze River Basin (left), and the corresponding loadings for each station (right). The**
**percentage variance explained is shown in the parenthesis.**




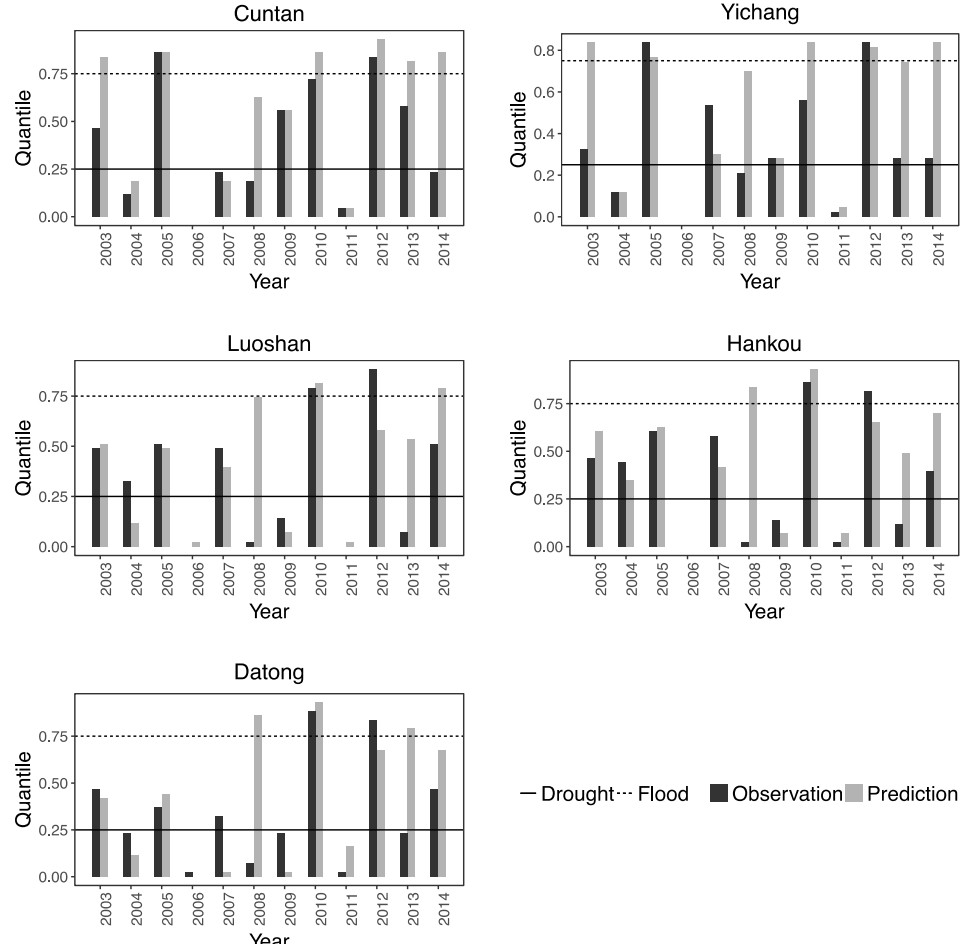


**Figure 8: Comparison of quantiles of streamflow between observations and predictions for each station. Based on observations during the pre-dam period, we obtained an ensemble of time series ranging from the largest to the smallest, then calculate the quantile of the streamflow relative to the time series. The dashed line denotes the flood and black line denotes the drought.**











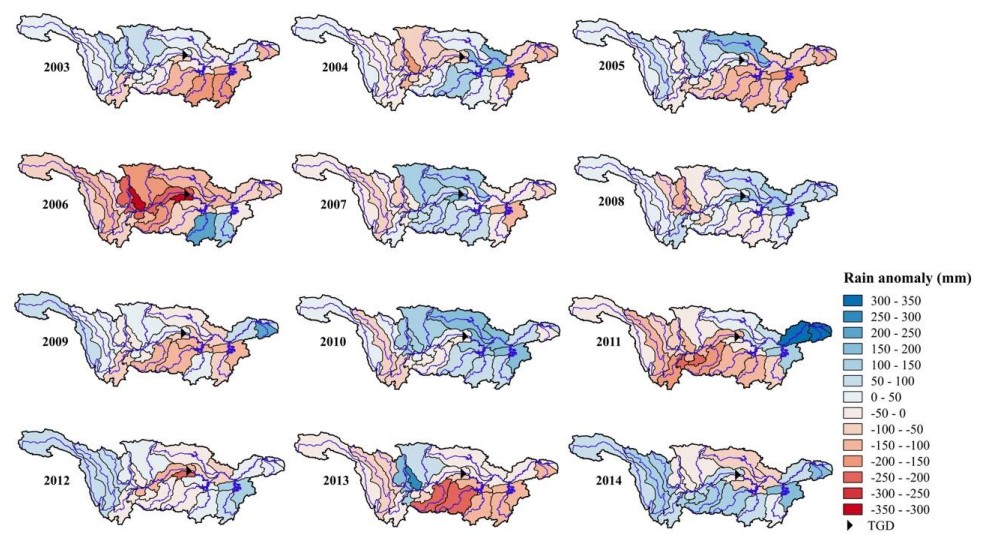


**Figure 9: The spatial and temporal distribution of rainfall anomalies over subbasins of Yangtze River Basin.**