# Peer review of "The effect of Three Gorges Dam and rainfall on summer"

_Hydrology and Earth System Sciences, 2017_

## Referee Comment (RC1) · Anonymous Referee #1 · 10 Apr 2017

The authors use canonical correlation analysis to develop linear model between streamflow and rainfall during pre-dam period (1960-2003), and the model is then used to predict streamflow during post-dam period (2003-2014). The difference between the prediction and the observation during the post-dam period is simply addressed to the dam effect. The method used by the authors is questionable, and the conclusions are not rigorous. I suggest to reject this paper for publication. The major deficiencies of this paper are:

1. The rainfall measurements across the whole Yangtze River basin were used to develop the regression models for five discharge stations, which is questionable. Since different discharge stations have different controlled upstream areas, it's more reasonable to use these rainfall stations located in the corresponding upstream area of each discharge station for the analysis. For instance, the authors presented in Lines 142-

144, "However, the rainfall in Subbasin Xiangjiang located at downstream of the dam has negative correlation with the streamflow, indicating that different climate events occurred over northwestern and southeastern part of Yangtze River Basin", I think the negative correlation is related to the fact that the rainfall in the downstream has not any contribution to the streamflow upstream.

2. The authors also divide the whole Yangtze River basin into 21 subbasins, while only data from 5 discharge stations are collected, which is far not enough for the analysis. I suggest to collect all the stremflow data from most discharge stations (at least stations represent the 21 subbasins) currently operated in the Yangtze River basin for the analysis.

3. The authors attribute the differences noted between the predicted and the measured streamflow during the post-dam period (e.g. Figure 6) to the dam effect, which is not rigorous. As seen from Figure 3, the correlation coefficients between the rainfall and streamflow are generally less than 0.7, indicating that only the rainfall cannot fully explain the measured streamflow dynamics. Figure 5 also shows that large uncertainties exist in the model performance. As such, simply attributing the difference to the dam effect is not correct without analyzing the impact of model uncertainties. In addition, the authors also note the lake effect, how will this affect the model development and prediction? The time lag between the rainfall and streamflow is also not considered in this study, how will this affect the analysis?

4. It's interesting to see from Figure 8 that the models predict totally different results for all the stations in comparison to the measurements for the year 2008, e.g. measured low flow vs. predicted flood, what's the reason for this? Is it related to the model deficiency? 5. The authors conclude in the abstract around Lines 28-30, "The comparison of risk type changes between observations and predictions during the post-dam period showed the dam effectively relieved the risk of upstream while not always played a positive role in risk mitigation of downstream", which sounds strange to me. It's surprise to see that the dam has minor effect on the downstream but affect the upstream area.

[Figure]

What's the reason for this? Is this conclusion still true if more discharge stations in the upstream area are included for the analysis?

Minor Comments: 1. Line 20: Is it correct to replace "each station" with "each sub-basin"?

2. Please add reference to the sentence "4 970 000 houses were flattened in the flood of year 1998 and agricultural disaster area was 33 900 km2 over upstream in the drought of year 2006" around Line 44.

3. Please rephrase the sentence "unclear parameter uncertainty and inaccurate physical process" around Line 61, it's not clear for "unclear parameter uncertainty", and what do you mean by "inaccurate physical process"?

4. Please rephrase the sentence "the uneven distribution of rainfall stations" around Line 62, I don't think this will "make it difficult for a single model to accurately characterize streamflow across the whole basin".

5. Please rephrase the sentence "But the use of highly correlated rainfall records among the adjacent gauges in a single model is problematic" around Line75, it's not clear.

6. Please also add the rainfall data to Figure 2.

7. Please rephrase the sentence "which are linearly interpolated using data from the closest two stations" around Line 110, it's not clear.

8. Can you add a table to introduce the name of each subbasin, it will be helpful to understand the text, for instance, presented around Lines 140-142, "the most significantly correlated subbasins are Jinshajiangupper, Yalongjiang and Jialingjiang".

9. Please rephrase the sentence "To solve this problem" around Line 156, what's the "problem"do you mean?

---

## Referee Comment (RC2) · Anonymous Referee #2 · 18 Apr 2017

This paper consider the relationship between rainfall from 136 weather stations and streamflow from 5 hydrological stations of Yangtze river, and then develop line models of two forecasting options with canonical correlation analysis. However, the runoff process of Yangtze river is a very complex process, should fully consider the respects of precipitation, evaporation, infilstration,underlying surface, groundwater and the control of reservoirs and lakes. It can't be simplified as a linear relationship. Besides that, the risk analysis of the dam running is not deep enough and not quantified. It is lack of theoretical analysis. The conclusions of the paper is not sufficient and reasonable. Therefore, I suggest to reject this paper.

---

## Author Comment (AC1) · 17 Jun 2017

We appreciate the suggestions and comments from the referee. This document discusses, point-by-point in response to the comments. Comment 1: The rainfall measurements across the whole Yangtze River basin were used to develop the regression models for five discharge stations, which is questionable. Since different discharge stations have different controlled upstream areas, it's more reasonable to use these rainfall stations located in the corresponding upstream area of each discharge station for the analysis. For instance, the authors presented in Lines 142-144, "However, the rainfall in Subbasin Xiangjiang located at the downstream of the dam has negative correlation with the streamflow, indicating that different climate events occurred over northwestern and southeastern part of Yangtze River Basin", I think the negative correlation is related to the fact that the rainfall in the downstream has not any contribution to the streamflow upstream. Response: The rainfall stations were selected to provide an approximately even spatial distribution to represent each sub-basin. Physically, the rainfall can directly contribute to the streamflow of the downstream stations. But, in this study, we applied the statistical approach to model the streamflow. Although rainfall in the Xiangjian subbasin does not contribute to streamflow upstream, it is indicative of the regional climate. The model does not intend to explicitly model the physical rainfall-runoff relationship, but uses a stochastic approach to incorporate information regarding regional climate relationships, that may be anti-correlated. The CCA process is used to maximize the correlation between the covariates and the target variable, so if one rainfall station is always negatively correlated, it is indicative of a regional climate that is negatively correlated to the basin streamflow of interest and the site is weighted appropriately.

Comment 2: The authors also divide the whole Yangtze River basin into 21subbasins, while only data from 5 discharge stations are collected, which is far not enough for the analysis. I suggest to collect all the streamflow data from most discharge stations (at least stations represent the 21 subbasins) currently operated in the Yangtze River basin for the analysis. Response: In this study, we choose the 5 important stations along the main stream with a view of demonstrating the method and capturing the changes of summer streamflow that was regulated by the dam. Because the 5 stations are the most important stations in Yangtze River and around big cities, they play an important role in flow monitoring and urban flood control. Cuntan is considered as the demarcation point of upstream and middle stream, and this is also the starting point of Three Gorges reservoir. The following stations are near the major cities or the gauging stations for monitoring the confluence of main stream and key tributaries. Second, because of the high summer flow in the Yangtze River, a small group of hydrological stations and the site's distribution can effectively help to observe the difference among the stations.

Comment 3: The authors attribute the differences noted between the predicted and the measured streamflow during the post-dam period (e.g. Figure 6) to the dam effect, which is not rigorous. As seen from Figure 3, the correlation coefficients between the rainfall and streamflow are generally less than 0.7, indicating that only the rainfall cannot fully explain the measured streamflow dynamics. Figure 5 also shows that large uncertainties exist in the model performance. As such, simply attributing the difference to the dam effect is not correct without analyzing the impact of model uncertainties. In addition, the authors also note the lake effect, how will this affect the model development and prediction? The time lag between the rainfall and streamflow is also not considered in this study, how will this affect the analysis? Response: The ability to study dam impacts on streamflow in this region is limited by the availability of data regarding dam releases and water withdrawals. Our approach was therefore to assess the relationship between rainfall and streamflow prior to and following the construction of the dam. The analysis is conducted using summer seasonal rainfall and streamflow data. In contrast to the lake effect, climate variability may likely drive the overall streamflow variability (Wang et al., 2017). The conveyance period for streamflow to travel from the upper catchment to the estuary is less than 3 months (about 14 days) (Chu et al., 2006). Given the relatively coarse temporal resolution considered, we did not consider a temporal lag in our model.

Chu, Z., Zhai, S., and Chen, X.: Changjiang River sediment delivering into the sea in response to water storage of Sanxia Rervoir in 2003, Acta Oceanologica Sinica, 25, 71-79, 2006. Wang, J., Sheng, Y., and Wada, Y.: Little impact of the Three Gorges Dam on recent decadal lake decline across China's Yangtze Plain, Water Resources Research, 53, 1-24, 2017.

Comment 4: It's interesting to see from Figure 8 that the models predict totally different results for all the stations in comparison to the measurements for the year 2008, e.g. measured low flow vs. predicted flood, what's the reason for this? Is it related to the model deficiency? Response: We would like to clarify that the Figure 8 does not the
predictions versus observations. We ran the model using rainfall during the post-dam period as the model covariates to produce the predictions of un-regulated streamflow, while the observations are the gauged streamflow during this period, which are impacted by the Three Gorges Dam post 2003. The difference between the modelled and the gauged values therefore represents the difference in streamflow as a result of dam operations in addition to model errors.

Comment 5: The authors conclude in the abstract around Lines 28-30, "The comparison of risk type changes between observations and predictions during the post-dam period showed the dam effectively relieved the risk of upstream while not always played a positive role in risk mitigation of downstream", which sounds strange to me. It's surprise to see that the dam has minor effect on the downstream but affect the upstream area. What's the reason for this? Is this conclusion still true if more discharge stations in the upstream area are included for the analysis? Response: The topography of the upstream is given priority to the hilly area with high altitude. The backwater can influence the streamflow of Cuntan, but imposes limited impact on streamflow of stations locating upstream of Cuntan, because Cuntan is the starting point of the forming reservoir by the dam, the backwater cannot flow far away along the channel. However, the downstream is given priority to the plain with many lakes (a total lake area of 15000 km2) (Wang et al., 2017). The water refilling and releasing can directly influence the inflow and water level of the downstream. For Cuntan station on the upstream, we think the difference between observations and predictions is due to the dam operation. While for the downstream, due to the interaction of river-lake, it cannot reflect the result directly impacted by the dam.

Wang, J., Sheng, Y., and Wada, Y.: Little impact of the Three Gorges Dam on recent decadal lake decline across China's Yangtze Plain, Water Resources Research, 53, 1-24, 2017.

For the minor comments, please see the supplement information, including all the response to the comments

Please also note the supplement to this comment:
http://www.hydrol-earth-syst-sci-discuss.net/hess-2017-159/hess-2017-159-AC1-
supplement.pdf

———————————————————

[Figure]

**Supplement:**

**Response to Referee 1**

**We appreciate the suggestions and comments from the referee. This document discusses, point-by-point in response to the comments.**

*Comment 1: The rainfall measurements across the whole Yangtze River basin were used to develop the regression models for five discharge stations, which is questionable. Since different discharge stations have different controlled upstream areas, it's more reasonable to use these rainfall stations located in the corresponding upstream area of each discharge station for the analysis. For instance, the authors presented in Lines 142-144, "However, the rainfall in Subbasin Xiangjiang located at the downstream of the dam has negative correlation with the streamflow, indicating that different climate events occurred over northwestern and southeastern part of Yangtze River Basin", I think the negative correlation is related to the fact that the rainfall in the downstream has not any contribution to the streamflow upstream.*

**Response:** The rainfall stations were selected to provide an approximately even spatial distribution to represent each sub-basin. Physically, the rainfall can directly contribute to the streamflow of the downstream stations. But, in this study, we applied the statistical approach to model the streamflow. Although rainfall in the Xiangjian subbasin does not contribute to streamflow upstream, it is indicative of the regional climate. The model does not intend to explicitly model the physical rainfall-runoff relationship, but uses a stochastic approach to incorporate information regarding regional climate relationships, that may be anti-correlated. The CCA process is used to maximize the correlation between the covariates and the target variable, so if one rainfall station is always negatively correlated, it is indicative of a regional climate that is negatively correlated to the basin streamflow of interest and the site is weighted appropriately.

*Comment 2: The authors also divide the whole Yangtze River basin into 21subbasins, while only data from 5 discharge stations are collected, which is far not enough for the analysis. I suggest to collect all the streamflow data from most discharge stations (at least stations represent the 21 subbasins) currently operated in the Yangtze River basin for the analysis.*

**Response:** In this study, we choose the 5 important stations along the main stream with a view of demonstrating the method and capturing the changes of summer streamflow that was regulated by the dam. Because the 5 stations are the most important stations in Yangtze River and around big cities, they play an important role in flow monitoring and urban flood control. Cuntan is considered as the demarcation point of upstream and middle stream, and this is also the starting point of Three Gorges reservoir. The following stations are near the major cities or the gauging stations for monitoring the confluence of main stream and key tributaries. Second, because of the high summer flow in the Yangtze

River, a small group of hydrological stations and the site's distribution can effectively help to observe the difference among the stations.

*Comment 3: The authors attribute the differences noted between the predicted and the measured streamflow during the post-dam period (e.g. Figure 6) to the dam effect, which is not rigorous. As seen from Figure 3, the correlation coefficients between the rainfall and streamflow are generally less than 0.7, indicating that only the rainfall cannot fully explain the measured streamflow dynamics. Figure 5 also shows that large uncertainties exist in the model performance. As such, simply attributing the difference to the dam effect is not correct without analyzing the impact of model uncertainties. In addition, the authors also note the lake effect, how will this affect the model development and prediction? The time lag between the rainfall and streamflow is also not considered in this study, how will this affect the analysis?*

**Response:** The ability to study dam impacts on streamflow in this region is limited by the availability of data regarding dam releases and water withdrawals. Our approach was therefore to assess the relationship between rainfall and streamflow prior to and following the construction of the dam. The analysis is conducted using summer seasonal rainfall and streamflow data. In contrast to the lake effect, climate variability may likely drive the overall streamflow variability (Wang et al., 2017). The conveyance period for streamflow to travel from the upper catchment to the estuary is less than 3 months (about 14 days) (Chu et al., 2006). Given the relatively coarse temporal resolution considered, we did not consider a temporal lag in our model.

Chu, Z., Zhai, S., and Chen, X.: Changjiang River sediment delivering into the sea in response to water storage of Sanxia Rervoir in 2003, Acta Oceanologica Sinica, 25, 71-79, 2006.

Wang, J., Sheng, Y., and Wada, Y.: Little impact of the Three Gorges Dam on recent decadal lake decline across China's Yangtze Plain, Water Resources Research, 53, 1-24, 2017.

*Comment 4: It's interesting to see from Figure 8 that the models predict totally different results for all the stations in comparison to the measurements for the year 2008, e.g. measured low flow vs. predicted flood, what's the reason for this? Is it related to the model deficiency?*

**Response**: We would like to clarify that the Figure 8 does not the predictions versus observations. We ran the model using rainfall during the post-dam period as the model covariates to produce the predictions of un-regulated streamflow, while the observations are the gauged streamflow during this period, which are impacted by the Three Gorges Dam post 2003. The difference between the modelled and the gauged

values therefore represents the difference in streamflow as a result of dam operations in addition to model errors.

*Comment 5: The authors conclude in the abstract around Lines 28-30, "The comparison of risk type changes between observations and predictions during the post-dam period showed the dam effectively relieved the risk of upstream while not always played a positive role in risk mitigation of downstream", which sounds strange to me. It's surprise to see that the dam has minor effect on the downstream but affect the upstream area. What's the reason for this? Is this conclusion still true if more discharge stations in the upstream area are included for the analysis?*

**Response:** The topography of the upstream is given priority to the hilly area with high altitude. The backwater can influence the streamflow of Cuntan, but imposes limited impact on streamflow of stations locating upstream of Cuntan, because Cuntan is the starting point of the forming reservoir by the dam, the backwater cannot flow far away along the channel. However, the downstream is given priority to the plain with many lakes (a total lake area of 15000 km$^2$) (Wang et al., 2017). The water refilling and releasing can directly influence the inflow and water level of the downstream. For Cuntan station on the upstream, we think the difference between observations and predictions is due to the dam operation. While for the downstream, due to the interaction of river-lake, it cannot reflect the result directly impacted by the dam.

Wang, J., Sheng, Y., and Wada, Y.: Little impact of the Three Gorges Dam on recent decadal lake decline across China's Yangtze Plain, Water Resources Research, 53, 1-24, 2017.

**Minor comments**

*Comment 1: Line 20: Is it correct to replace "each station" with "each subbasin"?*

**Response:** Yes, correct, it's a typing error. We will check the words again.

*Comment 2: Please add reference to the sentence "4 970 000 houses were flattened in the flood of year 1998 and agricultural disaster area was 33 900 km2 over upstream in the drought of year 2006" around Line 44.*

**Response:** Here, we showed an example for the previous sentence with references. The content is extracted from the cited papers listed. We will consider changing the position of the cited papers.

*Comment 3: Please rephrase the sentence "unclear parameter uncertainty and inaccurate physical*

*process" around Line 61, it's not clear for "unclear parameter uncertainty", and what do you mean by*

*"inaccurate physical process"?*

**Response:** Different hydrological models have their own advantages and disadvantages in the simulation of hydrological process. For example, in SWAT model, it divides the whole basin into many different sized hydrologic response units. For each unit, it has only one land use type, one soil type, one slope, et.al. The problem is that the exchange of water in different units is not considered in the model, thus, some physical process is inaccurate. We would like to direct the readers to refer to the references that we cited for details.

*Comment 4: Please rephrase the sentence "the uneven distribution of rainfall stations" around Line 62, I don't think this will "make it difficult for a single model to accurately characterize streamflow across the whole basin".*

**Response:** The used rainfall stations in this study are distributed in various parts of the basin, and the terrains include plateau, hilly area, plains and fluvial area. The altitude gradient in the west part is large and along with accumulated snow for years, while the altitude gradient in the east part is very small with heavy rainfalls. For different areas, we should apply different models to simulate the hydrological process based on the physical properties.

*Comment 5: Please rephrase the sentence "But the use of highly correlated rainfall records among the adjacent gauges in a single model is "problematic" around Line 75, it's not clear.*

**Response:** In the solving of multiple regression models, it requires the explanatory variables to be significantly independent with each other to get the unique solution. The existence of linear relationship among the explanatory variables may result in multicollinearity. This linearly dependent effect may make the regression coefficients unable to reflect the independent action and weaken the marginal effect of each other.

*Comment 6: Please add the rainfall data to Figure 2.*

**Response:** We have added the rainfall data to Figure 2 with blue line. Please see the following figure.

[Figure]

Figure 2: Monthly average streamflow (barplots) for each station and rainfall amount averaged over the

entire Yangtze River basin (Blue lines).

***Comment 7:*** *Please rephrase the sentence "which are linearly interpolated using data from the closest*

*two stations" around Line 110, it's not clear.*

**Response:** We got an ensemble of observed daily rainfall data for 136 weather stations with less than 5%

missing values from 1960 to 2014. For missing values in the long record for one specific station, we

firstly try to find the two closest stations without missing value on that day, then average the two values

from the two stations, we use this value to represent the missing value.

***Comment 8:*** *Can you add a table to introduce the name of each subbasin, it will be helpful to understand*

*the text, for instance, presented around Lines 140-142, "the most significantly correlated subbasins are*

*Jinshajiangupper, Yalongjiang and Jialingjiang".*

**Response:** The table was shown in the top of Figure 1 in the original manuscript, but was not clear

enough. We annotated each subbasin with a number in the map, and for each number, we specify the

subbasin name in the top rectangle. Please check it again.

***Comment 9:*** *Please rephrase the sentence "To solve this problem" around Line 156, what's the "problem"*

*do you mean?*

**Response:** The "problem" denotes the high correlations among the rainfall data for the adjacent stations.

One regression model requires the explanatory variables to be significantly independent with each other to avoid multicollinearity. Or else, this linearly dependent effect may make the regression coefficients unable to reflect the independent action.

---

## Author Comment (AC2) · 17 Jun 2017

We appreciate the suggestions and comments from the referee. We extract each comment from the overall comment. This document discusses, point-by-point in response to the comments. Comment 1: However, the runoff process of Yangtze river is a very complex process, should fully consider the respects of precipitation, evaporation, infiltration, underlying surface, groundwater and the control of reservoirs and lakes. Response: Yangtze River basin is the largest basin in China and subbasins within the Yangtze River Basin show a range of distinct properties and subsequently exhibit markedly different flow characteristics. Due to lacking sufficient documentation regarding data such as land cover, water levels above and below the dam, inflow and outflow of the lakes in the downstream, we simplify the problem in order to conduct

a suitable analysis given the data available. The analysis of dam influence is conducted along with the rainfall, excluding the non-studied factors such as land cover, urbanization, while the individual factor likely exerts a limited influence on the basin. From some related literatures (Wang et al., 2017), we can know that climate variability is the main driving factor of the runoff variability and already explains the observed inter-annual tendency. The integrated impact of TGD and other human activities may result in the difference between the predictions and observations, while for summer (JJA) flows during the post-dam period (12 years, from 2003-2014), the TGD-induced Yangtze flow changes can trigger instant alterations (Wang, 2013). For the above reasons, we emphasize the impact of rainfall and TGD on the summer river flow.

Wang, J.: Lake dynamics in the Yangtze Basin downstream of Three Gorges Dam driven by natural determinants and human activities, PhD, Dep. of Geogr., Univ. of Calif., Los Angeles, 2013. Wang, J., Sheng, Y., and Wada, Y.: Little impact of the Three Gorges Dam on recent decadal lake decline across China's Yangtze Plain, Water Resources Research, 53, 1-24, 2017.

Comment 2: It can't be simplified as a linear relationship. Response: In this study, we apply the canonical correlation analysis to explore the relationship between runoff and rainfall. To our best knowledge, the tendency of intra-annual rainfall is strongly consistent with that of runoff. Please also see the revised figure 1 and references (Zhang et al., 2011;Jiang et al., 2008). As the predominant driving factor of runoff, the rainfall in summer provides main water source to the Yangtze river. So, in the developed model, the log-transformed runoff is linearly correlated with the rainfall over subbasins, while that does not mean that the runoff of each individual hydrological station is a linear function of the rainfall. For the runoff in one specific station, it is not only simulated under the constraint of rainfall but also the constraint of observed runoff in other hydrological stations.

Figure 1: Monthly average streamflow (barplots) for each station and rainfall amount averaged over the entire Yangtze River basin (Blue lines). (Corresponding to the figure

2 in the original manuscript)

Jiang, T., Kundzewicz, Z. W., and Su, B.: Changes in monthly precipitation and flood hazard in the Yangtze River Basin, China, Int. J. Climatol., 28, 1471-1481, 2008. Zhang, Z. X., Chen, X., Xu, C. Y., Yuan, L. F., Yong, B., and Yan, S. F.: Evaluating the non-stationary relationship between precipitation and streamflow in nine major basins of China during the past 50 years, J. Hydrol., 409, 81-93, 2011.

Comment 3: The risk analysis of the dam running is not deep enough and not quantified. It is lack of theoretical analysis. Response: Thanks for pointing out this problem. We try to analyze the impact of spatial and temporal variability of rainfall on runoff during both pre-dam and post-dam period. Through the simple comparison, the TGD-induced runoff variation was then explicitly reflected. Since the rainfall is the main component of the high flow in flood seasons in summer, the use of rainfall anomaly can help to interpret the reason that the distinct difference between predictions and observations. Theoretically, our analysis may result in a conservative estimate of the runoff but provide an original comparison of the relative influence by the TGD, and this also considered the correlations among the runoff stations and rainfall stations over the entire Yangtze Basin.
* * *
[Figure]

**Fig. 1.**